# Lenalidomide Promotes Thrombosis Formation, but Does Not Affect Platelet Activation in Multiple Myeloma

**DOI:** 10.3390/ijms241814097

**Published:** 2023-09-14

**Authors:** Panpan Li, Bei Xu, Jiadai Xu, Yanyan Xu, Yawen Wang, Chen Chen, Peng Liu

**Affiliations:** 1Department of Hematology, Zhongshan Hospital, Fudan University, Shanghai 200032, China; 21211210008@m.fudan.edu.cn (P.L.); bbxu214@163.com (B.X.); xu.jiadai@zs-hospital.sh.cn (J.X.); 20111210138@fudan.edu.cn (Y.W.); chen.chen@zs-hospital.sh.cn (C.C.); 2Cancer Center, Zhongshan Hospital, Fudan University, Shanghai 200032, China; 3Department of Medical Oncology, Zhongshan Hospital, Fudan University, Shanghai 200032, China; 4Key Laboratory of Cell Differentiation and Apoptosis of Chinese Ministry of Education, Department of Biochemistry and Molecular Cell Biology, Shanghai Jiao Tong University School of Medicine, Shanghai 200032, China; xuyanyan901@shsmu.edu.cn

**Keywords:** multiple myeloma, immunomodulatory drugs, thrombosis, platelet

## Abstract

Lenalidomide, a well-established drug for the treatment of multiple myeloma, significantly enhances patients’ survival. Previous clinical studies have demonstrated that its main side effect is an increased risk of thrombotic events. However, the underlying mechanism remains unexplored. Therefore, this study aims to elucidate the mechanism and offer insights into the selection of clinical thrombotic prophylaxis drugs. Firstly, we conducted a retrospective analysis of clinical data from 169 newly diagnosed multiple myeloma patients who received lenalidomide. To confirm the impact of lenalidomide on thrombosis formation, FeCl_3_-induced thrombosis and deep venous thrombosis models in mice were established. To investigate the effects of lenalidomide on platelet function, both in vivo and in vitro experiments were designed. During the follow-up period, 8 patients developed thrombotic events, including 8 venous and 1 arterial. Further investigation using mice models demonstrated that lenalidomide significantly promoted the formation of venous thrombosis, consistent with clinical findings. To elucidate the underlying mechanism, assays were conducted to assess platelet function and coagulation. We observed that lenalidomide did not have any noticeable impact on platelet function, both in vitro and in vivo, while administration of lenalidomide resulted in significant decreases in prothrombin time, thrombin time, and prothrombin time ratio in patients, as well as a remarkable reduction in tail-bleeding time in mice. The administration of lenalidomide had no significant impact on platelet function, which may affect venous thrombus formation by affecting coagulation. Therefore, anticoagulant drugs may be superior to antiplatelet drugs in the selection of clinical thrombus prophylaxis.

## 1. Introduction

Multiple myeloma (MM), as the second largest hematologic malignancy, remains an incurable disease [1]. Lenalidomide, an immunomodulatory drug (IMiD), has been approved for use in China since 2013. The lenalidomide-based treatment regimen has significantly enhanced the clinical prognosis of patients with both newly diagnosed multiple myeloma (NDMM) and relapsed and refractory multiple myeloma (RRMM) [2]. Owing to its notable efficacy, oral administration, and convenience, lenalidomide is progressively establishing itself as the cornerstone of MM treatment. Extensive clinical data have consistently demonstrated an increased risk of thrombosis in MM patients taking lenalidomide [3,4,5]. In line with this, our center’s analysis of 931 NDMM cases further affirmed that IMiDs indeed serve as independent risk factors for thrombosis in MM [6]. However, despite the accumulation of clinical evidence, there remains a paucity of fundamental research investigating the mechanism through which lenalidomide elevates thrombotic risk.

Thrombosis, a complex process involving platelets, endothelial cells, and coagulation factors, plays a crucial role in vascular pathophysiology [7]. Within the bloodstream, vascular damage and extravasation events frequently occur. Under normal physiological conditions, hemostasis mechanisms maintain vascular integrity and regulate blood flow. However, pathological factors can disrupt the balance, leading to uncontrolled thrombus formation and subsequent occlusion of arteries or veins. Thrombotic events (TEs), such as arterial thrombosis events (ATEs) and venous thrombosis events (VTEs), significantly contribute to mortality among cancer patients [8,9]. Arterial thrombi, characterized by a high shear flow environment, primarily consist of platelets and typically form around ruptured atherosclerotic plaques. In contrast, venous thrombi are enriched with fibrin and red blood cells.

Considering the increased thrombotic risk associated with the use of IMiDs in MM patients, current clinical guidelines recommend the implementation of routine thromboprophylaxis for individuals receiving lenalidomide. However, there remains significant controversy regarding the optimal choice of drugs for this purpose. The International Myeloma Working Group (IMWG) suggests the use of aspirin, low molecular weight heparin (LMWH), and therapeutic doses of warfarin (INR 2.0–3.0) [4]. More recently, oral anticoagulants have been reported in the literature as capable of achieving comparable therapeutic outcomes [10]. These drugs can be further classified into anticoagulants and antiplatelet drugs. As of now, there is insufficient evidence to determine whether lenalidomide influences thrombotic risk through its impact on platelets or coagulation factors. Therefore, our research aims to offer novel insights into the selection of thrombosis prevention drugs.

In this study, the impact of lenalidomide on both arterial thrombosis and venous thrombosis was explored through a combination of clinical data analysis and in vivo and in vitro experiments. Moreover, the mechanisms underlying these effects were explored.

## 2. Results

### 2.1. Baseline Clinical Characteristics and Incidence of Thrombotic Events of the Enrolled NDMM Patients

From January 2013 to June 2021, 169 NDMM patients with VRD as the first-line treatment were included in this study. The baseline characteristics of the patients are described in detail in Table 1; the median age was 65 (32–87) years, and 61.5% of the sample are male patients. At the initial diagnosis, there were 51 (30.2%), 55 (32.5%), and 63 (37.3%) patients with International Staging System (ISS) in stages I, II, and III. IgG is the most common type of immunoglobulin. Furthermore, 62 (36.7%) patients had an Eastern Cooperative Oncology Group (ECOG) score greater than or equal to 2. Among all patients, 164 received thromboprophylactic medication, of which 157 (97%) received aspirin.

Our follow-up time was from the diagnosis of NDMM to one month after the end of first-line treatment. During this period, 8 patients had TEs. The characteristics of the TEs are described in detail in Table 2. One patient had deep vein thrombosis (DVT) and myocardial infarction (MI) at the same time. TEs included 8 VTEs and 1 ATE. There were 7 DVTs and 1 PE included in the 8 VTEs. Among the 8 patients suffering from thrombus, 2 patients did not receive thromboprophylaxis because of low platelet count, and the other 6 patients took aspirin, warfarin, and rivaroxaban.

### 2.2. Lenalidomide Administration Has Effect on Venous Thrombosis Formation in Mice In Vivo

To test whether lenalidomide affects thrombosis formation in vivo, lenalidomide (25 mg/kg, per day), lenalidomide (50 mg/kg, per day), or vehicle (DMSO) was administrated for 2 weeks to healthy male mice. FeCl_3_-induced carotid artery thrombosis and experimental DVT models were tested. Compared with the DMSO group, the lenalidomide 25 mg/kg group and lenalidomide 50 mg/kg group significantly affected the formation of venous thrombosis (Figure 1A–C), and there were significant statistical differences in both the length (*p* = 0.002; *p* = 0.0005) and weight (*p* < 0.0001; *p* < 0.0001) of thrombosis. In contrast, the formation of arterial thrombosis was not affected by lenalidomide administration in vivo (Figure 1D,E), the occlusion time of the experiment group and the control group did not show any differences, which is consistent with the low incidence of ATEs in clinical practice.

### 2.3. Lenalidomide Causes Changes in Platelet-Related Parameters in Patients but Does Not Affect Platelet Function In Vitro

In order to further explore the mechanism by which lenalidomide affects thrombosis formation, we first focused on whether lenalidomide affects platelet function. Firstly, we compared the changes in platelet related parameters between NDMM patients at initial diagnosis and before and after one cycle of VRD. Interestingly, some parameters showed significant changes, with no difference in platelet count (Figure 2A) before and after taking lenalidomide. Compared to baseline data, after taking one cycle of lenalidomide, the mean platelet volume (MPV, Figure 2B, *p* < 0.0001), platelet distribution width (PDW, Figure 2C, *p* < 0.0001), large platelet ratio (Figure 2D, *p* < 0.0001), platelet hematocrit (Figure 2E, *p* = 0.0005), and fibrinogen (Figure 2F, *p* = 0.0008) of patients increased significantly.

To evaluate the effect of lenalidomide on NDMM patients’ platelet activation in vitro, we pretreated the washed platelets from NDMM with lenalidomide (5 or 10 um) or DMSO for 30 min at 37 °C and performed platelet function assays (Figure 3A). Firstly, the aggregation of platelets was measured. Upon stimulation with thrombin (0.033 U/mL) and collagen (2 μg/mL; Figure 3B–E), the aggregation of platelets was measured and there was no difference between vehicle and lenalidomide. Secondly, we added platelets incubated with two different concentrations of lenalidomide or DMSO to human plasma for clot retraction testing. Similar to the above results, no significant differences were observed (Figure 3F). We recorded and analyzed the thrombus shrinkage rate of each group after 1 to 4 h of incubation, and no statistical differences were found (Figure 3G–J). Thirdly, the platelet spreading on immobilized fibrinogen in Tyrode buffer was performed (Figure 3L), which also showed no difference among the three groups (Figure 3K). Finally, we used flow cytometry to detect activation markers on the surface of platelets in both resting state and after thrombin stimulation (0.1 U/mL), and the results were consistent with before, with no statistical difference (Figure 3M–P).

### 2.4. Lenalidomide Administration Has No Effect on Platelet in Mice In Vivo

To clarify the effect of lenalidomide on platelet function and parameters in vivo, lenalidomide (25 mg/kg, once per day or 50 mg/kg, once per day) or DMSO were administrated for 2 weeks to mice (Figure 4A). The last dose was taken the night before narcotism, and platelet function was measured the next morning. We found that different from the changes in platelet related parameters in patients, in vivo data of mice showed no significant changes in platelet count (Appendix A), MPV (Appendix A), PDW (Appendix A), large platelet ratio (Appendix A), and platelet hematocrit (Appendix A) after taking lenalidomide.

Then, we prepared washed platelets from mice for functional experiments. There was no difference between the three groups, neither under stimulation of thrombin (0.033 U/mL, Figure 4B,C) or collagen (2 μg/mL, Figure 4D,E). Unsurprisingly, there was no significant change in the thrombus shrinkage rate between the experimental groups and the control group during the recorded four hours (Figure 4F–J). Similarly, lenalidomide does not affect platelet spreading on immobilized fibrinogen (Figure 4L,M). It is worth noting that compared to the control group, the tail bleeding time of the lenalidomide group was significantly shortened (Figure 4K, *p* < 0.0001; *p* = 0.0017), suggesting that taking lenalidomide affects tail bleeding time in mice. Exposure of P-selectin and JON/A with thrombin stimulation (0.1 U/mL) were detected and without difference between the three groups (Figure 4N–Q).

### 2.5. Lenalidomide Does Not Affect Healthy Donor’s Platelet Functions In Vitro

In order to further verify the effect of lenalidomide on platelet function in vitro, we collected blood samples from healthy contributors (Appendix A). Consistent with former aggregation tests, washed platelets were incubated with lenalidomide (5 µmol or 10 µmol) or DMSO for 30 min. As we predicted, platelet aggregation was not different among the groups in response to activating agent thrombin (0.033 U/mL, Appendix A) or collagen (2 μg/mL, Appendix A). These results demonstrated that lenalidomide does not affect platelet function in vitro. Clot retraction (Appendix A) and the platelet spreading (Appendix A) were estimated similarly and there were no differences among groups. Finally, we tested the expression of P-selectin and PAC1. Similar to the above results, no remarkable differences were observed (Appendix A).

### 2.6. Lenalidomide Affects Coagulation Parameters in NDMM Patients

In order to further explore the impact of lenalidomide on coagulation function in NDMM patients, we compared the coagulation parameters of NDMM patients who were enrolled at the time of diagnosis and after one cycle of taking lenalidomide. We found significant differences in prothrombin time (PT, Figure 5A, *p* < 0.0001), thrombin time (TT, Figure 5C, *p* < 0.0001), and prothrombin time ratio (PTR, Figure 5B, *p* < 0.0001), while there was no statistical difference in some changes in activated partial thrombin time (APTT, Figure 5D, *p* = 0.0881). These results may suggest that lenalidomide promotes the formation of venous thrombosis in vivo by influencing the coagulation pathway.

## 3. Discussion

The present study encompasses first an exploration of the underlying mechanism through which lenalidomide exerts its thrombotic promotion. Since the available clinical data exclusively associate lenalidomide with an augmented risk of thrombosis, one acknowledges the intricacy of the clinical milieu wherein patients seldom receive lenalidomide in isolation. Consequently, the primary objective of this study entailed an assessment of whether lenalidomide, in isolation, could indeed facilitate the genesis of arterial and venous thromboses in murine models. Remarkably, our findings reveal a marked proclivity of lenalidomide to enhance venous thrombosis formation. Subsequently, the underlying mechanism driving lenalidomide-mediated thrombosis was further explored. With prior literature underscoring the pivotal role of platelets in thrombosis, our first focus centered upon these blood constituents. To elucidate any potential impact, we evaluated platelet-associated parameters in 169 NDMM patients both pre-treatment and post-initial cycle of lenalidomide therapy. Significantly elevated MPV, PDW, large platelet ratio, platelet hematocrit, and fibrinogen levels were observed in these patients. In pursuit of a comprehensive understanding of lenalidomide’s effect on platelet function, both in vivo and in vitro experiments were designed. The in vitro experiments encompassed the utilization of washed platelets obtained from NDMM patients, who provided informed consent, wherein various concentrations of lenalidomide were applied prior to executing conventional platelet function assays. Notably, all functional assays conducted failed to demonstrate any discernible effect of lenalidomide on platelet function in NDMM patients. Given the potential confounding influence of the disease itself on platelet function in NDMM patients, we incorporated healthy volunteers to undergo in vitro experiments on platelet function, reaffirming the lack of lenalidomide-mediated impact on platelet function in vitro. It is widely recognized that in vivo circumstances often surpass the complexity of in vitro conditions. Consequently, a thorough analysis of lenalidomide’s potential effect on platelet function in vivo was carried out utilizing male mice. Following oral administration of lenalidomide for a duration of 14 days, blood samples were collected from the abdominal aorta to perform platelet function experiments. Intriguingly, lenalidomide consumption failed to elicit any change in platelet-related parameters or platelet function in mice. However, a rather surprising result emerged from the tail bleeding experiment, wherein lenalidomide was found to significantly abbreviate the bleeding time. To ascertain whether lenalidomide exerts its thrombotic promotion by influencing coagulation, pre- and post-treatment coagulation data were obtained and analyzed. The results demonstrated that compared to the pre-treatment phase, the TT, PT, and PTR were significantly reduced following lenalidomide administration, while the APTT exhibited no noteworthy changes. Based on the collective findings, one posits that lenalidomide seemingly lacks an impact on platelet function, but substantively promotes the in vivo formation of venous thrombosis by affecting the coagulation pathway.

The anti-MM effects of lenalidomide can be attributed to several underlying mechanisms. Upon binding to cereblon (CRBN) protein, lenalidomide triggers the activation of the CRBN E3 ubiquitination ligase complex, leading to the ubiquitination and subsequent degradation of transcription factors Ikaros (IKZF1) and Aiolos (IKZF3) [11]. This degradation process results in reduced expression of IRF4, Myc, and TNF-a, while increasing the expression of IL-2. These changes modify the functions of B and T cells and induce cytotoxic effects on MM cells. Additionally, lenalidomide possesses the ability to inhibit IκK and diminish IκB phosphorylation, thereby attenuating NF-κB activation and exerting anti-tumor effects [12]. In addition, lenalidomide demonstrates drug synergy by enhancing the sensitivity of tumor cells to dexamethasone and bortezomib [13]. Moreover, lenalidomide promotes the upregulation of CD38 by degrading Ikaros/Aiolos, thereby augmenting the daratumumab-mediated ADCC effect [14].

Thrombotic diseases encompass venous thrombosis and arterial thrombosis, both of which exert an impact on the prognosis of cancer patients and impose an elevated disease burden on individuals afflicted by cancer. Venous thromboembolism ranks as the second most common cause of mortality among cancer patients [15]. Risk factors for cancer-associated venous thromboembolism comprise tumor type, surgical interventions, chemotherapy, and employment of central venous catheters. In light of disease-related and treatment-related factors, MM patients face a heightened vulnerability to venous thromboembolism. A substantial cohort study conducted in Sweden demonstrated that patients diagnosed with undetermined monoclonal gammopathy (MGUS) and MM experience a 3.3-fold and 9.2-fold increase in the risk of DVT, respectively, relative to the general study population [16]. Our previous clinical study of 931 patients with multiple myeloma showed that “taking immunomodulatory inhibitors and renal insufficiency” were independent risk factors for thrombotic events in multiple myeloma patients with lenalidomide-based therapy [6]. Platelets, anuclear blood cells originating from megakaryocytes in the bone marrow, bear a close association with both physiological hemostasis and pathological thrombosis [17]. Platelets fulfil several crucial functions, including their adhesion to the site of endothelial damage and subendothelial collagen exposure in the circulation. This process entails platelet interaction with collagen and von Willebrand factor (vWF) through their glycoprotein GPVI and GPIb/la, eventually resulting in changes to platelet morphology [18]. Adhesion activates platelets and induces the secretion of ADP, 5-hydroxytryptamine, and thromboxane A2 (TxA2), thereby promoting platelet recruitment, activation, and the subsequent activation of aⅡbβ3 receptors within platelets [19]. Aggregation denotes the process by which platelets adhere to one another, occurring in two distinct phases: a rapidly reversible first phase and a more gradually irreversible second phase, during which early thrombosis formation transpires [7]. Aggregating agents bind to specific receptors on platelets, initiating intracellular signaling and activating aIIb β 3, which in turn binds to fibrinogen, instigating the “outward inward” signal transduction pathway. This event leads to the reorganization of cytoskeletal proteins within platelets, culminating in the formation of a stable thrombus [20]. Coagulation and fibrinolytic systems also assume pivotal roles in thrombus development. In this study, changes in laboratory tests related to coagulation were observed in NDMM patients undergoing lenalidomide treatment. PT serves as a screening test to detect exogenous coagulation factors, aiming to identify congenital or acquired deficiencies or inhibitors of fibrinogen, prothrombin, and coagulation factors V, VII, and X, as well as to monitor the dosing of oral anticoagulants. PT represents the preferred indicator for oral anticoagulant surveillance and may also be employed to assess liver protein synthesis function. Thromboembolic diseases and other pre-thrombotic conditions can experience a shortened clotting time. APTT serves as a screening test for endogenous coagulation factors and aids in confirming deficiencies of congenital or acquired coagulation factors VIII, IX, and XI, as well as the presence of their corresponding inhibitors. Moreover, APTT may be utilized to ascertain the deficiency of coagulation factor XII. Given the high sensitivity of APTT and heparin’s principal action on the endogenous coagulation pathway, APTT has emerged as the preferred indicator for monitoring standard heparin treatment. TT reflects the concentration of fibrinogen in plasma and the quantity of heparin-like substances present. Fibrinogen, otherwise referred to as coagulation factor I, represents the primary protein involved in the coagulation process.

In the present clinical practice, the administration of corresponding thromboprophylactic medications to most MM patients receiving lenalidomide treatment remains a subject of controversy regarding drug selection. Antithrombotics can be classified into three categories, namely antiplatelet agents, anticoagulants, and fibrinolytic solvents [21]. Presently, six types of antiplatelet drugs are available, including cyclooxygenase inhibitors (aspirin), P2Y12 antagonists (e.g., clopidogrel, prasugrel, and ticagrelor), glycoprotein IIb/IIIa inhibitors (e.g., abciximab, eptifibatide, and tirofiban), protease activated receptor 1 (PAR1) antagonists (e.g., vorapaxar and atopaxar), phosphodiesterase II inhibitor (cilostazol), and adenosine uptake blockade (dipyridamole) [7]. In addition, there are five types of anticoagulants—vitamin K antagonists (coumarin and acenocoumarol), antithrombin III activation (heparin and low molecular weight heparin), synthetic pentasaccharide inhibitors of factor Xa (fondaparinux and idraparinux), thrombin inhibitor (dabigatran), and factor Xa inhibitors (rivaroxaban, apixaban, edoxaban) [22]. Our study revealed that lenalidomide predominantly impacts coagulation, thereby promoting the occurrence of venous thrombosis within the body. These findings may indicate that anticoagulants could yield more favorable outcomes in preventing thrombotic events during clinical practice [23].

In conclusion, despite certain limitations, for the first time, this study unveils a novel discovery regarding the effect of lenalidomide on thrombosis formation in vivo. Remarkably, lenalidomide exhibits a notable propensity to enhance venous thrombosis formation, while displaying no significant effect on arterial thrombosis formation. Moreover, for the first time, this study has demonstrated lenalidomide does not affect platelet function in both in vivo and in vitro, rather, it affects coagulation-related indicators. In this light, these findings offer valuable insights when selecting thromboprophylactic drugs for patients receiving lenalidomide in clinical practice.

## 4. Materials and Methods

### 4.1. Patients and Clinical Information

The medical records of 169 consecutive NDMM patients treated with VRD (bortezomib 1.3 mg/m^2^, subcutaneously, day 1, 8, 15; lenalidomide 25 mg, orally, days 1–14; dexamethasone 40 mg, oral, day 1, 8, 15, 22) in Zhongshan Hospital of Fudan University between January 2013 and June 2021 were retrospectively reviewed. Under the terms of informed permission, platelet- and coagulation-related parameters from patients at baseline and after one cycle of VRD administration were collected. TEs were defined by the International Society on Thrombosis and Hemostasis [24]. The diagnosis of TEs were verified by objective imaging techniques, including venous ultrasonography, venography, and computed tomographic pulmonary angiography. This study adhered to the World Medical Association’s Declaration of Helsinki and was approved by our hospital’s Ethics Committee (B2017-031R).

### 4.2. Experimental Design

For the in vitro experiments, platelets of healthy volunteers and NDMM patients before receiving chemotherapy were isolated from venous blood under informed consent. All NDMM patients were recruited from inpatients in the Hematology Department of Zhongshan Hospital, Fudan University, Shanghai, China. Patients experiencing infections or thrombotic diseases, or those receiving antithrombotic therapy were excluded. Platelets that were isolated from healthy volunteers and NDMM patients were incubated with lenalidomide or DMSO for 30 min, and platelet functional assays were performed to evaluate the effect of lenalidomide on platelet function. The healthy mice were orally administered lenalidomide or vehicle to confirm the impact of the drug on platelets in vivo.

### 4.3. Animals

C57BL/6J male mice were purchased from Biocytogen Pharmaceuticals, Jiangsu, China. All animal procedures met the standard of the Guide for the Care and Use of Laboratory Animals published by the US National Institutes of Health (publication No. 85-23, revised 1996) and were approved by the Animal Care Committee of Zhongshan Hospital, Fudan University. Only male mice were studied to avoid the potential impact of hormones on platelets [25,26,27].

### 4.4. Drug Administration

For drug preparation, 375 mg lenalidomide (catalog No.191732-72-6; MedChemExpress, Shanghai, China) was dissolved in dimethyl sulfoxide (DMSO) and 0.9% NaCl to 25 mg/mL, storing at −20 °C for no more than 1 months. This stock solution was further diluted with PEG300, 0.9% NaCl, and Tween-80 to 2.5 mg/mL and 5 mg/mL before use and orally administered within 20 min. The healthy male mice (8–10 weeks old) received oral lenalidomide at 25 mg/kg, 50 mg/kg, or an equal volume of vehicle (DMSO) daily for 2 weeks before experimental evaluation. The selected lenalidomide dose was based on previous literature [28] and the treatment duration was selected because it is the physiological life span of platelets [29].

### 4.5. Complete Blood and Washed Platelet Counts

A mouse orbital blood sample (50 uL) was collected into a heparinized tube (BDMicrotainer) and stored in a constant temperature water bath at 37 °C. Before testing, the blood sample was diluted with an equal volume of physiological saline preheated to 37 °C. Platelet and red blood cell counts, hemoglobin levels, and hematocrit were analyzed on an automated hematology analyzer (SysmexXT-2000i). For the washed platelet counts, one volume of platelets was diluted with 9 volumes of modified Tyrode solution, and the final measurement was the average of 3 repeated tests.

### 4.6. Platelet Preparation and Aggregation

Washed platelets were prepared as described previously [30]. In short, blood was collected from the abdominal aorta of mice anesthetized with isoflurane and placed in a syringe containing 100 μL/mL White anticoagulant (2.94% sodium citrate and 136 mmol/L glucose (pH 6.4)). Then, 0.1 μg/mL PGE1 (prostaglandin E1; catalog number 745-65-3; Sigma Aldrich, St. Louis, MO, USA; 1:10,000) and 1 U/mL apyrase (catalog number 9000-95-7; Sigma Aldrich; 1:2000) were added in the samples. The samples were diluted with an equal volume of physiological saline preheated to 37 °C, and then centrifuged at 1050 rpm for 10 min. The platelet-rich plasma was transferred to a new centrifuge tube and centrifuge at 2100 rpm for 10 min. The supernatant was discarded and the platelets resuspended in Tyrode solution (1 mM Ca^2+^, 1 mM Mg^2+^, pH 7.4). Washed platelets were adjusted to a density of 3 × 10^8^ platelets/mL and aggregation of platelets was stimulated in response to collagen and thrombin as previously described [31].

### 4.7. Platelet Spreading and Clot Retraction

Assays of platelet spreading on fibrinogen (50 ug/mL) were performed as previously described [31,32]. The adherent platelets were stained with rhodamine conjugated phalloidin and images were captured under fluorescence microscopy (×100 objective). The spreading area of platelets was quantified using the ImageJ software (version 1.47). For clot retraction, we added 100 μL washed platelets and 300 μL human platelet-depleted plasma and 0.4 U thrombin to the final concentration of 1 U/mL. The mixture was then incubated at 37 °C. Photos were taken at different time intervals and the data were analyzed with ImageJ software (version 1.47).

### 4.8. Flow Cytometry

To evaluate the expression of platelet P-selectin and αIIbβ3 upon activation, washed platelets were adjusted with Tyrode solution to 3 × 10^7^ platelets/mL. The diluted washed platelets were then stained with 10 μg/mL fluorescein isothiocyanate–conjugated anti-human CD62P or 5 μg/mL fluorescein isothiocyanate conjugated PAC1 for human platelets (10 μg/mL PE-conjugated anti-mouse CD62P or 50 μg/mL PE-conjugated anti-mouse JON/A for mouse platelets). Thrombin was then added or not to the final exhibited concentration. The mixture was incubated at 37 °C for 30 min and then the platelets were fixed with 400 μL 4% paraformaldehyde. To assess the effect of lenalidomide on platelet activation, the expression of P-selectin and αIIbβ3 was quantified on a CytoFLEX S Flow Cytometer (Beckman Coulter Life Sciences, Indianapolis, IN, USA).

### 4.9. FeCl_3_-Induced Thrombosis Model and Tail Bleeding

Ferric chloride (FeCl_3_)-induced carotid artery injury murine thrombosis models were performed as previously described [32]. In short, we anesthetized the mice with isoflurane and exposed the carotid arteries. A 2-mm piece of filter paper wet with FeCl_3_ was used to injure the vascular for 3 min. A tail-bleeding time experiment was implemented as previously described. In brief, 8-week-old mice were anesthetized with isoflurane and their tails were prewarmed in 0.9% isotonic saline at 37 °C for 5 min. A fraction from the tail tip was snipped (3 mm) and the injured tail was immediately soaked in warm saline again. The bleeding time was recorded from the start of bleeding to cessation of the bleeding [32].

### 4.10. Experimental DVT Model

The DVT models of mice were generated as described previously [33]. In short, the mice were anesthetized with isoflurane. Their intestines were gently pulled out with a cotton swab and placed on sterile gauze moistened with warm physiological saline for protection. We carefully separated the inferior vena cava from the aorta using a cotton swab, and then used a 7-0 suture to pass through the inferior vena cava. A 30G needle (diameter of 0.31 mm) was placed above the inferior vena cava, and then a suture was used to ligate it before gently removing the needle. All visible inferior vena cava side branches were also sutured. After the surgery was completed, the intestines were returned to the abdomen and the abdomen closed completely. The mice were placed on a heating blanket until they awakened. Thrombosis was collected after 48 h.

### 4.11. Statistical Analysis

Statistical analyses were performed with GraphPad Prism 8.0. Continuous variables were presented as the mean ± standard error of the mean (SEM). The unpaired *t* test and paired *t* test were used for comparisons between two conditions. One-way ANOVA was performed for multiple comparisons. Values of *p* < 0.05 were considered statistically significant.

## Figures and Tables

**Figure 1 ijms-24-14097-f001:**
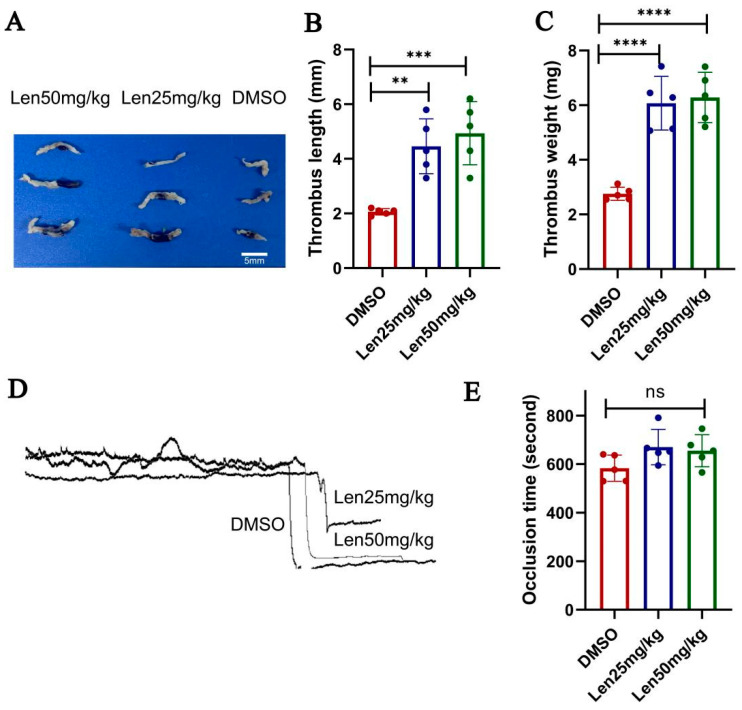
The effect of lenalidomide on thrombosis formation in mice. (**A**–**C**) Lenalidomide significantly affected the formation of venous thrombosis. (**A**) the photo of venous thrombosis; (**B**) thrombosis length (n = 5); (**C**) thrombosis weight (n = 5). (**D**,**E**) lenalidomide had no effect on the formation of arterial thrombosis. (**D**) occlusion time (n = 5), ns, no significance; (**E**) blood flow records of FeCl_3_-induced carotid artery thrombosis formation in mice. ** *p* < 0.01, *** *p* < 0.001, **** *p* < 0.0001.

**Figure 2 ijms-24-14097-f002:**
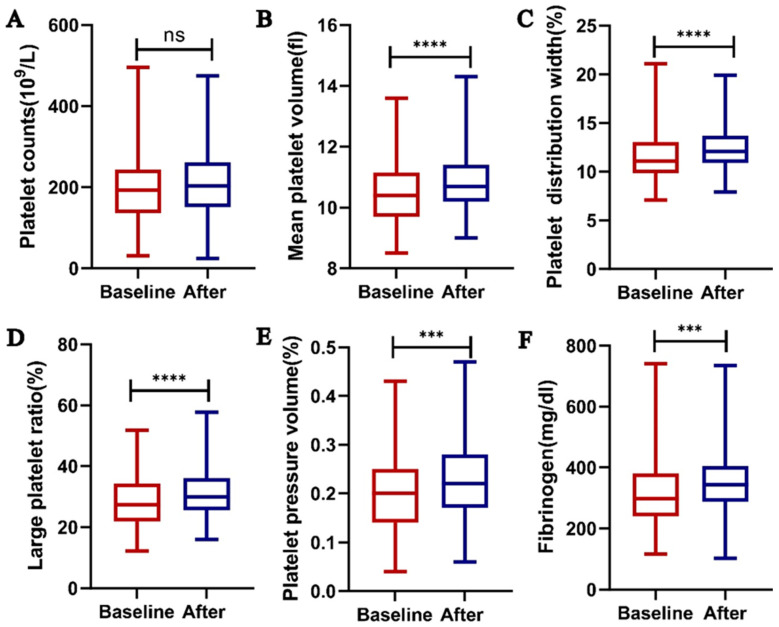
Changes in platelet-related parameters in patients after one cycle of taking lenalidomide. (**A**) platelet counts (n = 169, ns); (**B**) mean platelet volume (n = 169, *p* < 0.0001); (**C**) platelet distribution width (n = 169, *p* < 0.0001); (**D**) large platelet ratio (n = 169, *p* < 0.0001); (**E**) plateletocrit (n = 169, *p* = 0.0005); (**F**) fibrinogen (n = 169, *p* = 0.0005). *** *p* < 0.001, **** *p* <0.0001.

**Figure 3 ijms-24-14097-f003:**
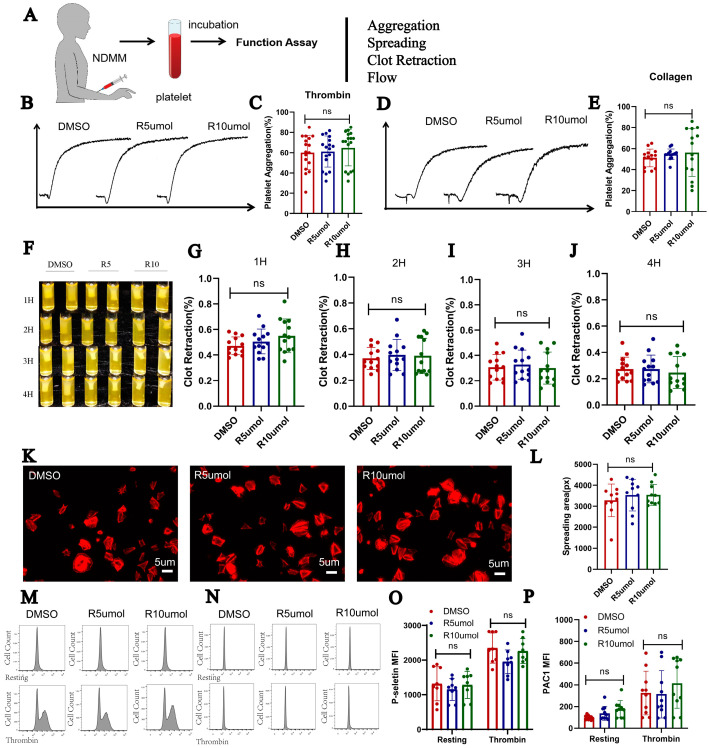
Lenalidomide did not affect platelet activity in NDMM patients in vitro. (**A**) Design of experiment. (**B**) Platelet aggregation curves when platelets were stimulated with thrombin (0.03 U/mL). (**C**) Maximum platelet aggregation of platelets after stimulation with thrombin (n = 17). (**D**,**E**) Platelet aggregation stimulated with collagen (2 μg/mL) (n = 14). (**F**–**J**) results of clot retraction. Bar plots show mean ± SD of clot retraction levels from 3 independent experiments (n = 13). (**K**,**L**) Spreading of the platelets incubated with DMSO and lenalidomide (n = 10). (**M**–**P**) Exposure of P-selectin (n = 8) and PAC1 tested by flow cytometry (n = 10).

**Figure 4 ijms-24-14097-f004:**
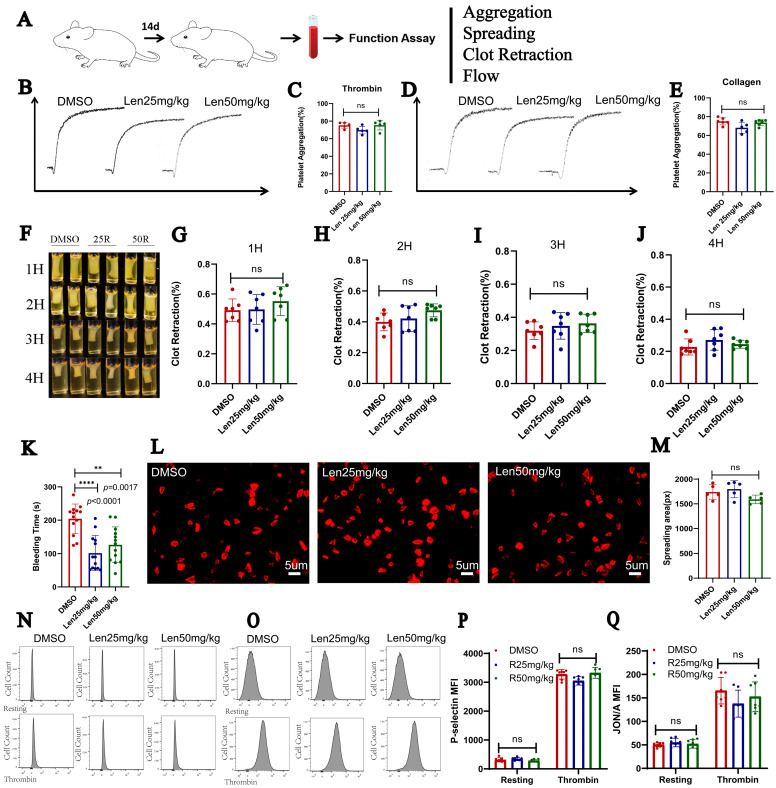
Lenalidomide did not affect platelet activity in mice in vivo. (**A**) Design of experiment. (**B**) Platelet aggregation curves stimulated with thrombin (n = 5) (0.03 U/mL). (**C**) Maximum platelet aggregation of platelets after stimulation with thrombin. (**D**,**E**) Platelet aggregation stimulated with collagen (n = 6) (2 μg/mL). (**F**–**J**) results of clot retraction (n = 7). (**K**) The tail-bleeding time of mice (n = 13). (**L**,**M**) Spreading of the platelets incubated with DMSO and lenalidomide (n = 5). (**N**–**Q**) Exposure of P-selectin (n = 7) and JON/A (n = 7) tested by flow cytometry. ** *p* <0.01, **** *p* <0.0001.

**Figure 5 ijms-24-14097-f005:**
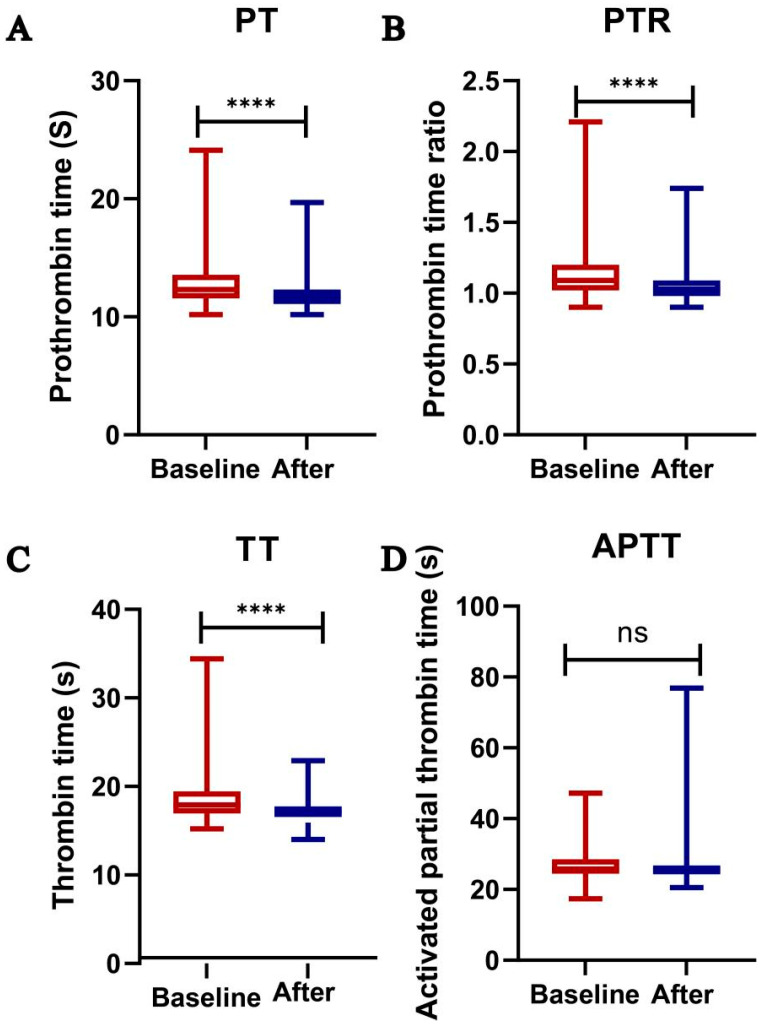
Changes in coagulation-related parameters in patients after one cycle of taking lenalidomide. (**A**) prothrombin time (n = 169, *p* < 0.0001); (**B**) prothrombin time ratio (n = 169, *p* < 0.0001); (**C**) thrombosis time (n = 169, *p* < 0.0001); (**D**) activated partial thrombin time (n = 169, *p* = 0.0881). **** *p* < 0.0001.

**Table 1 ijms-24-14097-t001:** Baseline clinical characteristics of 169 NDMM patients.

	Level	Number	Percent (%)
**Number**		169	
**Age (median (range))**		65	32–87
**Age**	**<65**	81	47.9
	**≥65**	88	52.1
**Gender**	**Female**	65	38.5
	**Male**	104	61.5
**ISS**	I	51	30.2
	II	55	32.5
	III	63	37.3
**RISS**	I	15	8.9
	II	111	65.7
	III	43	25.4
**IFE type**	**IgG**	90	53.3
	**IgA**	48	28.4
	**IgD**	8	4.7
	**Light chain**	18	10.6
	**Non-secretor**	5	3
**Anticoagulation therapy**		164	97
	**ASA**	157	95.7
	**Rivaroxaban**	3	1.8
	**Clopidogrel**	1	0.6
	**Warfarin**	3	1.8
**Cytogenetics**		150	88.8
	**Del17p**	15	10
	**t (4;14)**	27	18.1
	**t (14;16)**	2	1.2
	**t (11;14)**	24	16.1
	**Gain1q21**	85	57
**ECOG (%)**	**0–1**	107	63.3
	**≥2**	62	36.7

Abbreviations: *ISS*, International Staging System; *RISS*, Revised International Staging System; *IFE*, immunofixation electrophoresis; *ECOG,* Eastern Cooperative Oncology Group.

**Table 2 ijms-24-14097-t002:** Description of thrombosis events during follow-up.

Case	Gender	Age	Isstage	TE	Ecog	Prophylaxis
1	Male	75	1	R DVT	0	ASA
2	Male	53	3	R DVT	2	NA
3	Male	81	1	L DVT	1	ASA
4	Male	70	3	R DVT/MI	1	Warfarin
5	Female	56	3	PE	3	NA
6	Male	66	2	B DVT	1	ASA
7	Male	52	2	B DVT	1	ASA
8	Male	66	3	R DVT	1	Rivaroxaban

Abbreviations: *L*, left; *R*, right; *B*, both legs; *DVT*, deep vein thrombosis; *PE*, pulmonary embolism; *MI,* myocardial infarction; *ASA*, aspirin; *NA*, without prophylaxis.

## Data Availability

Data supporting the findings of this study are available from the corresponding author upon reasonable request.

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
