# Peer review of "Lenalidomide Promotes Thrombosis Formation, but Does Not Affect Platelet Activation in Multiple Myeloma"

_ijms, 2023, doi:10.3390/ijms241814097_

Round 1

Reviewer 1 Report

The article entitled “The Mechanism Underlying Lenalidomide-Induced Thrombosis Susceptibility” talks about a clinical important topic such as the thrombotic risk of 169 patients with newly diagnosed multiple myeloma (NDMM) on treatment with lenalidomide. The authors conducted an in vivo and in vitro experimental and retrospective study using a appropriate methods and materials (platelets incubated with 25 mg of lenalidomide that have been isolated from healthy subjects and NDMM patients; platelet from healthy male mice receiving 25 mg of lenalidomide; measurement of P-selectin and αIIbβ3 as markers of platelet function; PT, TT, and PTR as markers of coagulation). The results of this study  show that the lenalidomide causes venous thrombosis through the coagulation activation and it does not affect the platelets. On this basis, the authors suggest the use of antithrombotic treatment with anticoagulant. I have no any comment to make. Therefore, I think that this article is suitable for publication in its current version.

Reviewer 2 Report

The authors used patients's samples and animal model to study the thrombosis mechanism of lenalidomide treatment in myeloma patients.

1. Though interesting with support from animal models, there is few new information from the study.

2. For patients's characteristics, was there any information from the patients which were also increase the risks of patients, such as hyperlipidemia, diabetics, old age? 

3. Since it is a retrospective study, how the authors had baseline samples for incubation study?

3.  The figure 2B, 2C, and 2D are very similar, are them real from the raw data or mis-use? And 2B, 2C, 2D, 2E, 2F are marked as very significant power, but the figure are not persuaded from the figure. 

4. Same situation could be seen in  figure 5A, 5B, and 5C, the figures seems not persuaded with significant power. 

5. On the study of figure 3 and figure 4, the authors should mention of samples' size for more information.

Round 2

Reviewer 2 Report

The authors modified the manuscript by reviewers' comments. I think it is acceptable to be published in the journal.